Regeneration dynamics of Portulacaria afra in restored succulent thicket of South Africa

Galuszynski Nicholas C. nicholas.galuszynski@gmail.com
Spekboom Thicket research group, Botany Department, Nelson Mandela University , Gqeberha , South Africa
Ferrenberg Scott
Electronic publication date: 2023 May 1
Publication date: 2023
Volume: 11
Electronic Location ID: e15081
Received 2022 Sep 13; Accepted 2023 Feb 25
Copyright: ©2023 Galuszynski
Copyright year: 2023
Copyright holder: Galuszynski
License: This is an open access article distributed under the terms of the Creative Commons Attribution License, which permits unrestricted use, distribution, reproduction and adaptation in any medium and for any purpose provided that it is properly attributed. For attribution, the original author(s), title, publication source (PeerJ) and either DOI or URL of the article must be cited.
License URL: https://creativecommons.org/licenses/by/4.0/

Keywords: Spekboom, Seedling recruitment, Ecological processes, Ecosystem functioning, Albany Subtropical Thicket, State transitions

Funding: South African National Biodiversity Institute This work was funded by the South African National Biodiversity Institute as a postdoctoral grant. The funders had no role in study design, data collection and analysis, decision to publish, or preparation of the manuscript.

==============================
Introduction

Over-grazing by livestock has resulted in the widespread degradation of South Africa’s succulent thicket ecosystems. This is characterised by a significant reduction in the cover of the dominant succulent shrub, Portulacaria afra. Because this species is unable to regenerate naturally in degraded habitat, active reintroduction is required to restore ecosystem function. However, reintroduction success is relatively low, and the recruitment barriers for this species are poorly understood.

Methods

By conducting pairwise plot surveys in actively restored and adjacent degraded succulent thicket habitats, the extent of P. afra seedling abundance in these contrasting ecosystem conditions is quantified.

Results

Seedling abundance was significantly greater in restored ecosystems (W = 23, p = 0.03225). Additionally, seedlings found in restored habitats were strongly associated with open habitat, whereas seedlings in degraded ecosystems were more restricted to nurse sites (X2 = 122.84, df = 2, p-value < 2.2e–16). A weak (R2 = 0,237), but significant (p = 0, 0295) correlation between P. afra cover and seedling abundance was recorded.

Conclusion

Active restoration of succulent thicket habitat through P. afra reintroduction appears to overcome recruitment barriers. This may suggest that, despite the poor survival of introduced individuals, natural recruitment could contribute to the regeneration of restored succulent thicket ecosystems.

Introduction

The delicate balance of biotic and abiotic interactions that maintain the persistence of arid/semi-arid (dryland) ecosystems is particularly vulnerable to disruption, making these systems prone to transitioning into a degraded state (Verwijmeren et al., 2013; Aronson et al., 1993). Once degraded, drylands often lack the resilience to naturally regenerate and may require active restoration (Shackelford et al., 2021; Aronson et al., 1993). The ecological success of these restoration efforts is frequently evaluated based on the return of vegetation structure, species diversity, and indicators of ecosystem functioning (Wortley, Hero & Howes, 2013). Here, the recruitment dynamics of Portulacaria afra Jacq., the target plant species used in succulent thicket restoration, is compared between restoration plots and adjacent degraded habitat. The data presented here contribute to the growing evidence that succulent thicket restoration facilitates the return of ecosystem functioning.

Portulacaria afra has long been considered a valuable browse plant in the Eastern Cape of South Africa (Oakes, 1973), supporting the wool and mohair industries over the past three centuries (Beinart, 2008). This prolonged livestock production has led to widespread overgrazing of succulent thicket vegetation, often resulting in the near complete loss of P. afra cover. This has shifted the ecosystem from a dense evergreen woodland to a savanna-like vegetation dominated by an ephemeral understorey and bare ground (Lechmere-Oertel, Kerley & Cowling, 2005). This change in vegetation structure represents a transition that the ecosystem is unlikely to recover from unless actively revegetated (Lechmere-Oertel, Kerley & Cowling, 2005). Succulent thicket restoration, therefore, focuses predominantly on the reintroduction of P. afra. The feasibility of this intervention has been tested at scale in a biome-wide experiment, consisting of approximately 330 Thicket-Wide restoration Plots (TWPs) established across the distribution range of succulent thicket between the years 2008 and 2009 (Mills & Robson, 2017), which form the basis for the adhoc experiment presented here.

The ecological importance of P. afra in intact succulent thicket has been well described. This species is capable of facilitating landscape-scaled processes by modifying soil properties (van Luijk et al., 2013; Lechmere-Oertel et al., 2008; Mills & Fey, 2004) and providing the micro-climate required for recruitment of woody canopy species (Wilman et al., 2014; Sigwela et al., 2009). Many of these properties are reported to return in restored ecosystems, including the accumulation of soil organic carbon (Schagen et al., 2021; Mills & Cowling, 2014; van der Vyver et al., 2013) and the recovery of some recruitment dynamics (van der Vyver et al., 2013). The remainder of thicket restoration success has largely been quantified in terms of living P. afra individuals present in restoration plots, measured to evaluate different planting treatments implemented in the TWPs (van der Vyver, Mills & Cowling, 2021a; Mills & Robson, 2017). With an average of 28% of individuals surviving in large-scale plantings conducted over a 12-year period (Mills & Robson, 2017), this approach would suggest that succulent thicket restoration has a relatively low success rate. Measuring individual survival, however, may under-represent the return of ecological processes and cannot detect whether an ecosystem is capable of regenerating without further intervention.

The current perspective on P. afra seedling recruitment is that it is rare and sporadic (Sigwela et al., 2009; Midgley & Cowling, 1993), likely requiring nurse plant microsites (Adie & Yeaton, 2013). However, these studies were conducted in predominantly intact thicket ecosystems, and no attempts have been made to evaluate the extent of seedling recruitment in the patchy cover of restored thicket habitat (Fig. 1). The data presented here (a snapshot of P. afra seedling abundance) suggest that while P. afra recruitment may be sporadic, seedlings do establish readily in restoration plots but are absent in adjacent degraded habitat. Thus, indicating a return of natural regeneration dynamics.

Figure 1 An example of a Thicket Wide Plot (TWP) and the distribution of the ten TWPs surveyed for Portulacaria afra seedlings.

The paired sampling quadrats are indicated with red dashed lines. Note the three distinct vegetation conditions, restored P. afra cover in the TWP in the foreground, the open savanna-like habitat (degraded thicket) surrounding it and the densely vegetated hills in the background (intact thicket). The spatial distribution of the ten TWP surveyed is provided in the top right inset (Maps data: Google, SIO, U.S. Navy, NGA, GEBCO), which is contextualised as a red polygon in relation to South Africa to the left of the inset.

Methods

Study sites

Of the 330 TWPs, ten were selected for seedling surveys. These ten plots are representative of the ecological range of succulent thicket, including sites located near the northern, southern, and eastern limits of the vegetation’s distribution (Fig. 1). Some TWPs were mistakenly established near valley bottoms in an adjacent, frost-prone habitat not suitable for P. afra (Duker et al., 2015; Duker et al., 2020), while others were subject to livestock browsing after being established (van der Vyver et al., 2021b). Therefore, not all TWPs were suitable for inclusion in this study, and any plots with reported survival below 20% were excluded from selection. The ten TWPs included in this study all occurred within a matrix of degraded succulent thicket and had some degree of intact fencing, however, evidence of herbivory was present in many of the plots (signs of browsing and dung). Approval for field surveys was acquired verbally from all landowners (S. Nelson, H. Steyn, L. Wright, A. Botha, H. Bekker, T. Murray, M. Wienard, D. van der Decke, and C. Comely) prior to conducting the surveys.

As the TWPs were not established with the intention of testing the contribution of revegetation to seedling recruitment, the adjacent degraded habitat did not include any fenced-off control. This study, therefore, represents an ad hoc experiment, where the degraded control sites reflect a range of different management practices, including stocking of livestock and/or indigenous game.

Seedling surveys

All surveys were conducted in July 2021, following a severe multi-year regional drought (Archer et al., 2022), and 12–13 years after the initial P. afra reintroduction. The full extent of each TWP (50 × 50 m) was surveyed and at each sampling plot, P. afra cover was visually estimated (Table 1). All seedlings encountered were documented, and their relative position in reference to a possible nurse shrub was noted. Seedlings were defined as individuals that were either obvious seedlings that had recently germinated or exhibited a typical “mat” above-ground morphology and red pigmentation (Fig. 2). Seedlings were either located directly below a nurse shrub, on the edge of a nurse shrub canopy, or in an open habitat defined as bare ground and thus, not associated with any nurse plants. For each TWP a 50 ×50 m quadrat was marked out in the adjacent degraded thicket habitat (Fig. 1). The full extent of these quadrats were surveyed, and all seedlings were documented in the same manner as before. The number of seedlings detected in degraded thicket plots was insufficient to draw conclusions on the tendency for P. afra recruitment to be associated with nurse plants outside of the fenced-off TWPs (Fig. 3). Therefore, two additional 50 × 50 m quadrats were surveyed in a moderately degraded succulent thicket habitat (with an estimated 30% P. afra canopy cover) identified by a landowner. These two quadrats covered the majority of the accessible area described by the landowner and were located in a portion of the property managed for indigenous game hunting.

The ten TWPs surveyed in this study were distributed across a range of succulent thicket community types. The understorey plant community was, therefore, highly variable across locations, and no trend for specific nurse species could be evaluated. Rather, each site had its own suite of nurse shrubs, predominantly taking the form of dwarf shrubs originating from neighbouring habitats.

Data analysis

Differences in seedling abundance between restored and degraded succulent thicket was determined via Wilcoxon rank sum tests for non-parametric data (Hollander & Wolfe, 1973). Differences in the likelihood of seedlings being located in nurse sites between restored and degraded thicket were determined from Pearson’s Chi-squared tests of independence (Hope, 1968). The correlation between seedling abundance and P. afra cover was evaluated using a general linear model (Wilkinson & Rogers, 1973). It was evident that site-specific management of the degraded habitat had no effect on seedling recruitment, as seedlings were rare in all degraded sites, no additional analyses were conducted to account for management practices. All data were analysed using base R, version 4.2.0 (R Core Team, 2022).

Results and Discussion

Overcoming barriers to unaided recruitment of target species is a major challenge in ecosystem restoration (Fick et al., 2016; Acácio et al., 2007; Standish et al., 2007). Lack of plant density (seed sources) coupled with predation, increased seed loss by wind and rain, and the loss of suitable germination sites promote recruitment failure from a reduced seed pool (Volis, 2019). The decline in ecosystem condition is well described in degraded succulent thicket habitat. The loss of the P. afra canopy exposes soils to the elements; leading to the loss of organic content (Lechmere-Oertel et al., 2008; Mills & Fey, 2004), reduced water holding capacity (Mills & de Wet, 2019; van Luijk et al., 2013), increased light exposure, and extreme fluctuations in soil temperature (Lechmere-Oertel et al., 2008). This could potentially eliminate suitable germination sites for P. afra, disrupting natural regeneration.

Table 1 Portulacaria afra canopy cover and seedling abundacne recorded in each 50 × 50 m quadrat.

All degraded sites were exposed to some degree of herbivory (either livestock and/or native browsers) whereas the restored sites were fenced off to limit browsing. Some evidence of browsing was however evident in many of the restoration plots (i.e., animal paths and dung), but to a far lesser degree than in the degraded sites.

	Degraded	Restored	
Plot ID	P. afra cover	Seedlings (count)	P. afra cover	Seedlings (count)	
p0008	10	2	7	1	
p0014	0	0	60	311	
p0032	5	4	55	84	
p0035	0	0	90	0	
p0058	0	0	45	3	
p0089	0	0	35	7	
s0015	0	0	40	63	
s0045	0	0	10	0	
s0081	0	1	10	10	
s0093	0	0	10	0	

Figure 2 Portulacaria afra seedling morphology.

(A) A reecntly germinated P. afra seedling (likely from within the past year), note the distinct red pigmentation. (B) The primary stem tends to exhibit lateral growth to create a hooked morphology as the seedling establishes. (C, D) This lateral growth is continued by side branches to form a mat before initiating vertical growth. Note that all the morphologies depicted here were considered seedlings in the surveys as they had not matured to a point that they exhibit vertical growth or the loss of the red leaf pigmentation typical of juvinile P. afra. Any individuals with green leaves or vertical branches were not considered seedlings.

Figure 3 Box-plot comparing seedling abundance between restored and degraded succulent thicket habitat.

Despite the harsh abiotic conditions present in degraded thicket habitat, P. afra seed does exhibit the ability to germinate (Fig. 3). Seedlings of various growth stages (Fig. 2) were detected in degraded and restored habitat, suggesting that seedlings can establish once germinated. The lack of recruitment in degraded thicket may, therefore, be best described as a consequence of seed limitation due to a lack of plant cover, rather than a lack of suitable germination sites. This is supported by a weak (R2 = 0, 237), but significant (p = 0, 0295) correlation between P. afra cover and seedling abundance, and the fact that seedlings were detected in degraded succulent thicket habitat with limited cover (Table 1). The active reintroduction of P. afra appears to overcome this seed limitation. More seedlings (W = 23, p = 0.03225, Fig. 3) were found in restoration plots (total seedlings in all TWPs = 477, mean = 47.90 seedlings/plot, SD = 97.20, DF = 9) than in adjacent degraded habitat (total seedlings in paired degraded plots = 7, mean = 0.7 seedlings/plot, SD = 1.34, DF = 9). Further work is, however, required to better describe the interplay between canopy cover, herbivory, and recruitment, as sites with high canopy cover (e.g., 90 %) did not appear to support seedlings (Table 1).

Seedling microhabitat frequency differed between degraded and restored plots (X-squared = 122.84, df = 2, p-value < 2.2e–16). Seedlings found in fenced-off restoration plots (N = 477) tended to occur in open, exposed sites (69.2%) more frequently than elsewhere (9% found on the edge of shrub canopies and 21.8% below a nurse shrub). Seedlings established in degraded thicket (N = 186, including the additional two targeted quadrats) were found 49.5% of the time beneath a canopy of nurse shrubs, 28.5% of the time on the edge of a canopy of nurse shrubs, and 22% of the time in the open.

Recruitment of plant propagules is strongly associated with intact succulent thicket habitat that supports dense stands of P. afra (Sigwela et al., 2009). The cool, moist microclimate created by dense vegetation cover in this dryland ecosystem is postulated to mimic the subtropical conditions in which much of the thicket canopy tree species originated, facilitating germination and recruitment (Wilman et al., 2014). The loss of canopy cover, therefore, halts broad community assembly processes in degraded succulent thicket, evident in the general lack of seedlings reported in degraded thicket (Sigwela et al., 2009). Restoring this canopy by planting P. afra appears to reverse this loss of ecological functioning by regenerating soil organic carbon (Schagen et al., 2021; van der Vyver et al., 2013; Mills & Cowling, 2006) and providing a cool, shaded microclimate. This has facilitated the return of natural recruitment of woody canopy species within 35 years of P. afra reintroduction (van der Vyver et al., 2013) and the regeneration of soil bacterial community compositions within 12 years of P. afra reintroduction (Schagen et al., 2021). In the ad hoc experiment presented here, significant returns of P. afra recruitment were detected 12–13 years after restoration intervention. Thus, this further contributes to the growing body of evidence that the active reintroduction of P. afra is a suitable intervention for the restoration of degraded succulent thicket ecosystems. However, while the presence of P. afra is required to produce seed for recruitment, unlike other thicket species, canopy cover and nurse shrubs do not appear to facilitate the germination of this ecologically significant species.

Open sites are hotter (Lechmere-Oertel et al., 2008) and drier (Adie & Yeaton, 2013) than below plant canopies, yet P. afra seedlings appear capable of germinating in these sites in degraded and restored habitats, even during a prolonged and severe drought (Archer et al., 2022). This is likely enabled by multiple adaptations. Firstly, P. afra can shift between C3 and CAM photosynthetic pathways, ensuring productivity during hot, dry conditions (Guralnick & Gladsky, 2017; Ting & Hanscom, 1977). Furthermore, seedlings exhibit a red pigmentation (Fig. 2), likely due to high levels of anthocyanins, which may contribute to tolerating aridity, UV radiation, and extreme temperature fluctuations (Chalker-Scott, 1999).

As the open sites are comparable between restored and degraded plots, both are composed of bare soil, lack organic matter, and are exposed to direct sunlight, the different spatial patterns of seedling recruitment described for restored and degraded habitats are unlikely a consequence of abiotic conditions. Rather, the association of P. afra seedlings with nurse shrubs, argued to be part of a cyclic succession process by Adie & Yeaton (2013), is more likely due to herbivory protection in open and actively browsed habitats. The sampling sites in this study all supported livestock and/or indigenous game (reduced in TWPs due to fencing), whereas the study site in Adie & Yeaton (2013) supported livestock in a relatively intact succulent thicket. Thus, the tendency for P. afra seedlings to be associated with nurse shrubs is unlikely to be indicative of landscape-wide community assembly patterns, as suggested by Adie & Yeaton (2013), but rather, a consequence of herbivore density.

Herbivory can act as a barrier to the recruitment of palatable target species in restoration initiatives (MacDougall & Wilson, 2007; Opperman & Merenlender, 2000). Herbivory was found to be one of the key determinants of reintroduced P. afra survival in the TWPs that provided the basis for this study (van der Vyver et al., 2021b). In cases where the fenced enclosures were removed (either intentionally by landowners or accidentally by large animals), P. afra reintroduction success was markedly reduced, with no plants surviving in some instances (van der Vyver et al., 2021b). It should, however, be noted that the restoration plot with the second highest seedling density (84 seedlings recorded) exhibited clear signs of browsing and the fence had been pushed down by cattle in recent years (i.e., the reintroduced P. afra were well established before being exposed to browsing by livestock). This could imply that once established, restored succulent thickets can withstand some browsing without compromising local regeneration patterns, but the carrying capacity of restored succulent thicket must be closely monitored.

Conclusion

One of the major challenges facing the restoration of dry ecosystems is overcoming barriers to seedling establishment. The restoration of succulent thicket ecosystems relies predominantly on the reintroduction of the target plant species P. afra, which exhibits rare and sporadic recruitment from seed. This intervention appears to ameliorate seed set limitations, promoting the recruitment of this species in restored habitats. No evidence of nurse plants facilitating seedling establishment was detected in fenced restoration plots; however, seedlings were commonly associated with nurse plants in degraded sites exposed to greater levels of herbivory. Nurse shrubs do not, therefore, appear to be tied to P. afra recruitment as previously suggested. To facilitate the return of recruitment dynamics in restored succulent thicket, herbivore pressure should be reduced and/or restoration should target ecosystems with low herbivore densities.

Supplemental Information

Supplemental Information 1 R Script and associated P. afra seedling count data

The Wilcoxon rank sum test and Pearson’s Chi-squared test of independence are coded into a single R Script. The relevant data sets for these tests are provided in two spreadsheets.

Click here for additional data file.

The author would like to thank Anisha Dayaram, Andrew Skowno and Alastair Potts for supporting the collection of data presented in this study, Shayla Tricam for assisting in field sampling, and two anonymous reviewers whose comments greatly improved the quality of this manuscript.

Additional Information and Declarations

Competing Interests

Author Contributions

Field Study Permissions

Data Availability

The author declare that they have no competing interests.

Nicholas C. Galuszynski conceived and designed the experiments, performed the experiments, analyzed the data, prepared figures and/or tables, authored or reviewed drafts of the article, and approved the final draft.

The following information was supplied relating to field study approvals (i.e., approving body and any reference numbers):

The relavent landowners (Shane Nelson, Hendrik Steyn, Leonard Wright, Andries Botha, Hansie Bekker, Tim Murray, Mark Wienard, Dongi van der Decke, and Chad Comely) approved the seedling surveys and provided access to field sites before any data collection was conducted.

The following information was supplied regarding data availability:

Counts of seedlings associated with different micro-habitats in restored and degraded thicket ecosystems are available in the Supplemental Files.

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
