# Peer review of "Regeneration dynamics of Portulacaria afra in restored succulent thicket of South Africa"

_PeerJ, doi:10.7717/peerj.15081_

## Round 0.1 · original submission · Major Revisions

Please pay particular attention to the reviewers' comments regarding potential issues with the experimental design. Clarifying the design and issues that it may present for data analysis and interpretation can improve the manuscript.

Reviewer 1 ·

Basic reporting

The main purpose of this manuscript is to compare seedling recruitment of the perennial succulent shrub Portulacaria afra between areas that have undergone active restoration efforts and nearby degraded habitats. The author reports on the location of these seedlings as occurring either under existing shrub canopies, shrub drip lines, or interspace, which makes this ecologically interesting. Especially in the presence of active grazing pressure. In general, I do believe this is reportable data that could be relevant to managers working on restoration.
While the author has stated the objective clearly in the first paragraph of the introduction, I struggled with the organization and clarity of the manuscript. The English is ok, although there are grammatical errors throughout the manuscript. The results and discussion sections have been combined into a single narrative. I believe this has created an organizational challenge to manuscript in it’s current state, and important components of both the methods and results are missing.
The author does supply a long list of citations in regards to P. afra restoration, specifically efforts related to this study. However, the manuscript can be greatly improved by more thoroughly explaining the relevance of the previous restoration efforts and studies and how they are directly related to the study design of this current manuscript.

Experimental design

There are some key components that should be addressed to contextualize the experimental design. Otherwise, it is difficult to assess whether this study design is valid or not.

The methods state that TWPs were surveyed and all P. afra seedlings were recorded. However, they do not tell us when these plots were surveyed (season?year?), how many times they were surveyed, or how much time has passed since restoration efforts were taken.

The author should also detail, what they mean by "recruitment" In figure 2 they show us differences in seedling life stages. Were all of these stages recorded as seedlings? I am not familiar with the growth rates of P. afra, but I would caution the author to not assume 100% survivorship for all seedlings to make it to the recruitment stage. As reported in the manuscript, this is a single snapshot of germination. Not recruitment.

In the introduction the author states that mean survivorship of planted P. afra in the restoration plots was 28% as reported by Mills and Robson 2017. Mills and Robson (2017) also report a large amount of variation in survivorship across the TWPs. This begs some immediate questions for the author of this manuscript:
• What were the densities of adult P. afra in both the degraded and restoration plots at the start of this study? These would be the primary seed source for germinants within the plot, although we can think of other ways seeds may arrive.
• In a raw comparison between restoration and degraded plots this presents a challenge, if there are different densities of parent plants. How does the author account for differences in parent shrubs? As of now, I struggle with the circular argument that we find more seedlings in plots that have more propugal pressure (i.e., seeds available to germinate).
• Related to potential differences in adult P. afra densities, the restoration plots also include an exclosure to keep out grazers. So in a sense, there are 2 treatments at play: 1) the planting of P. afra and whatever potential differences in density exist; 2) grazing and no grazing. In the current study design of comparing paired restored and degraded plots, these things are not accounted for. But, they can be by utilizing a mixed model approach to account for starting density differences, grazing exclosure differences, and blocked site differences.
Out of the 330 TWPs, the author states that they selected 10 that are “are representative of the ecological range of succulent thicket, including sites located near the dry interior and wetter coastal distribution limits of the vegetation” Lines 77-79. This implies that there is an environmental component inherent to the study design, particularly one related to site level moisture and aridity. However, the author does not assess differences across the plot locations. See my blocked site comment above.

Validity of the findings

As the data are currently presented, it is difficult to assess. The difference in germination microhabitat between restoration plots and degraded plots is compelling. However, unequal sampling design needs to be accounted for in order to have a more rigorous assessment. See comments above.

Additional comments

Specific Comments:

Lines 51-54: As it is written, it is not clear if the 330 TWPs are in reference to your own study. It reads as if you are citing older work, but not your actual work

Line 52: "approximately 330" Why not report the exact number.

Lines 55-57: Check grammar. These sentences can be re-written for clarity

Line 64: After how many years did they record the mean survivorship of 28%?

Line 72-74: This is a statement for the discussion

Lines 80-83: Restoration efforts were at locations not suitable for P afra? Survivorship and stand density are different than suitable habitat. If the plots had 20% survivorship of planted P. afra, is not the habitat suitable for those 20%? Your reasoning "avoid instances of plantings being undertaken in habitats unsuitable for P. afra" Were not the plantings of P. afra and not other species? This is not clear.
This statement is also contradictory to the second sentence of this paragraph stating that plots are representative of the range of environmental conditions.

Line 89: Are the dominant nurse shrubs P. afra? Or something else. Please describe in site descriptions. Degraded and restored plots. If nurse shrubs are different species, this should also be included in study desing.


Line 130: "Seedlings established in degraded thicket (N=186) " 186 is not what is reported in Fig 3, and is different than what is reported in line 120 (9).

Reviewer 2 ·

Basic reporting

Line 51: The feasibility of which has been tested…
I would change to: The feasibility of this intervention has been tested…

Experimental design

Lines 129-133: I presume this is including seedlings in degraded areas but not limited to the paired degraded sites ie. also additional sites elsewhere to include a sufficient sample size? How was the area determined in which this data was collected? Perhaps elaborate more in lines 95-96 and explain how this was quantified. Were all degraded sites included for seedling sampling subject to herbivory or were any fenced/herbivory excluded?

Validity of the findings

Presuming that no degraded areas were surveyed where herbivory is excluded, this is a pity since it would have been a valuable finding to determine if these habitats could recover with herbivore exclusion alone or if habitat is sufficiently propagule-limited that active restoration of Portulacaria afra is necessary to provide sufficient seed to allow for further seedling establishment. Although the limited numbers of seedlings even where protected by nurse plants would suggest that active restoration is still necessary to create a viable seed source within a local area. Perhaps this point could be discussed in some more detail, linked to the previous note?

Additional comments

Overall a well written and interesting article, with important findings for conservation and restoration of this ecosystem.

---

## Round 0.2 · accepted · Accept

I appreciate your efforts to provide thoughtful and detailed responses to the reviewers' comments. I assessed the revision and feel that the manuscript has been improved and is ready for publication.